# OpenReview forum: "GPTBIAS: A Comprehensive Framework for Evaluating Bias in Large Language Models"
_ICLR.cc/2025/Conference — ICLR 2025 Conference Withdrawn Submission_

### Official Review · Reviewer_aMi6 · 2024-10-31

**Soundness:** 2
**Presentation:** 4
**Contribution:** 1
**Rating:** 5
**Confidence:** 4

**Summary:**

This paper proposes GPTBIAS to flexibly evaluate different types of biases in Large Language Models. The framework is a 4-step process: instruction generation, LLM response collection, bias evaluation, and score calculation. Compared to previous evaluation measures, GPTBIAS does not require labels, sample pairs, or access to weights.

**Strengths:**

- The paper is well-written and easy to follow.
- The motivation and the approach is clear.
- The method has the greatest flexibility in terms of evaluation requirements.

**Weaknesses:**

- My main concern is the lack of technical novelty in this approach. The framework largely relies on the evaluation of GPT-4 (ChatGPT), which is already commonly done in practice. Although I appreciate the formalism here, I think the contribution to the LLM community is limited.
- While the authors provided an extensive list of evaluation tables, that itself doesn't prove the reliability of GPTBIAS. Due to the heterogeneous nature of StereoSet and CrowS-Pairs, it is difficult to directly compare them with it. That being said, it would also be helpful to see some ablation studies on the components of the evaluator, and demonstrate how that impacts evaluation.
- If I understand correctly, GPTBIAS relies on bias attack instructions that are generated by an LLM. If so, I am uncertain about the stability of this method. It would be interesting to see how the evaluation score varies for different evaluation runs. That is, how stable is the evaluation when run multiple times?
- (minor) it would be nice to have citations in brackets - use \citep

**Questions:**

See Weaknesses.

---

### Official Review · Reviewer_voXb · 2024-11-04

**Soundness:** 2
**Presentation:** 2
**Contribution:** 2
**Rating:** 3
**Confidence:** 4

**Summary:**

The paper presents a novel framework, GPTBIAS, designed to evaluate biases in large language models. Unlike traditional bias evaluation methods that require access to a model's intermediate outputs, GPTBIAS uses advanced LLMs like GPT-4 to evaluate bias without model access constraints. This framework introduces "Bias Attack Instructions," which are crafted prompts targeting nine specific bias types, including gender, race, and socioeconomic status, to evaluate model responses comprehensively. GPTBIAS not only generates a quantitative bias score but also provides detailed insights into bias types, affected demographics, reasons behind biases, and suggestions for reducing these biases. Through experiments on popular LLMs, the authors demonstrate that GPTBIAS effectively captures both overt and subtle biases, underscoring the importance of thorough bias evaluation in LLM development and deployment​

**Strengths:**

1. The bias of the large language models is important and interesting.

2. The authors propose a framework that evaluates nine different biases.

**Weaknesses:**

1. The paper is not well-written. It is hard to read and logical problems are many.  For example, in lines 13-15, since "different constraints" are mentioned, the author should list more than just one constraint. In line 68, the phrase "that can a wide range of bias types" is missing a verb. In lines 246-247, "the corresponding language model" should be revised to "the response of the corresponding language model." Please ask for some writing help before submission.

2. The paper claims that GPTBIAS offers more than just a bias score by providing additional insights, including bias types, affected demographics, underlying reasons for biases, and suggestions for improvement. However, the experiments focus primarily on the bias score and bias types (Tables 3, 4, and 5), and do not include experiments demonstrating the value or impact of the other additional information. It would be beneficial for the authors to present a case study that analyzes how these insights aid in evaluating bias or what important conclusions can be drawn from them.

3. The comparison between GPTBIAS and baseline metrics, StereoSet and CrowS-Pairs, is unclear, as shown in Table 3. Since GPTBIAS scores range from 0 to 1, while baseline scores are typically greater than 1, as shown in Table 1, the differing evaluation scales make it challenging to interpret the performance gap between the methods. To clarify this comparison, the authors could either normalize the comparison by evaluating GPTBIAS using the same metric as the baselines or propose a new, unified metric suitable for all these methods."

4. Lines 370-372 suggest that baseline methods perform worse because current LLMs can avoid generating biased predictions on their test datasets. This raises an important question: is the new test dataset used in GPTBIAS the primary factor behind its apparent superiority? To address this, an ablation study comparing the evaluation frameworks on the same test dataset would be valuable.

5. The target LLMs used in the experiments—LLaMA, OPT, BLOOM, and GPT-3—seem somewhat outdated. The results would be more
compelling if the experiments were conducted on more recent models, such as LLaMA-3 and GPT-4. It would be helpful if the authors could clarify whether these recent models are not suitable for GPTBIAS. If GPTBIAS can be applied to newer models, a discussion on their performance under GPTBIAS would be highly valuable.

6. The analysis of intersectional bias (Section 5.3) and various bias types (Section 5.4) is too brief. The authors provide only a brief description of the data in Table 4 and Table 5, without drawing further conclusions from these results. To strengthen the discussion, the authors could elaborate on how intersectional bias and the statistics for different types of bias contribute to the overall analysis of model bias or inform strategies for mitigating bias in models.

**Questions:**

Check the weakness parts.

**Details Of Ethics Concerns:**

This work talks about the bias analysis which will bring the potential concerns.

---

### Official Review · Reviewer_rtb3 · 2024-11-04

**Soundness:** 2
**Presentation:** 3
**Contribution:** 2
**Rating:** 5
**Confidence:** 3

**Summary:**

The paper presents GPTBIAS, a framework developed to comprehensively assess biases in large language models (LLMs) by leveraging GPT-4 as a meta-evaluator. The framework generates "Bias Attack Instructions" to probe responses from various LLMs across nine bias categories, including gender, race, and socioeconomic status, and computes an intersectional bias score that highlights overlapping biases affecting multiple demographic groups. GPTBIAS provides quantitative bias scores and qualitative insights, distinguishing it from traditional methods like StereoSet and CrowS-Pairs. Results from experiments on models such as BLOOM, LLaMA, and GPT-3.5 demonstrate GPTBIAS's ability to capture nuanced biases, which are often missed by existing evaluation metrics.

**Strengths:**

- GPTBIAS introduces an innovative approach by employing GPT-4 as an evaluator to conduct bias analysis on other LLMs. This design broadens the scope of bias evaluation, providing a unique angle for analyzing how biases interact across demographic dimensions.

- The paper demonstrates methodological rigor through detailed steps in bias instruction generation, model response collection, and response analysis.

- The overall structure of the framework is clearly described.

- The actionable feedback generated by GPTBIAS is a valuable asset for developers and researchers aiming to refine AI models.

**Weaknesses:**

- The framework relies exclusively on GPT-4 as the evaluator, raising concerns regarding accessibility and potential bias propagation from GPT-4 itself. Teams without access to GPT-4 due to cost or licensing restrictions may find it challenging to implement GPTBIAS effectively. Exploring alternative evaluators, such as GPT-3.5 or open-source LLMs, could increase GPTBIAS’s accessibility and applicability. Testing and discussing whether these alternatives provide similar evaluative accuracy could be a valuable addition.

- The paper introduces intersectional bias scores but lacks concrete guidance on how developers can practically use these scores to improve LLMs. Without examples or a case study, the interpretability and actionable value of these scores remain unclear.

- The current framework applies only to English, limiting its relevance for models trained in or applied to other languages. Biases manifest differently across languages, meaning the framework may overlook key insights when assessing non-English models. The paper would benefit from discussing plans or modifications necessary to extend GPTBIAS to multilingual models, such as adapting bias instructions for cultural and linguistic variations. Even hypothetical steps toward multilingual adaptation would broaden the framework's appeal and utility.

- Given the detailed structure of the bias instructions and reliance on GPT-4, GPTBIAS appears computationally intensive, especially for large-scale models like BLOOM and LLaMA. This may make it impractical for frequent evaluations or use by teams with limited computational resources. Proposing a streamlined or resource-efficient version of GPTBIAS, perhaps by limiting the scope of bias types or reducing instruction complexity, would make the framework more accessible. A comparison between the full framework and a lighter version could illustrate how well each captures biases.

**Questions:**

1. Could the authors provide an example or further guidance on how model developers can use intersectional bias scores to inform improvements?

2. Given the reliance on GPT-4, could the authors test GPTBIAS with GPT-3.5 or other accessible models to assess performance differences and possible trade-offs?

3. Are there plans to expand GPTBIAS for non-English LLMs? What modifications might be necessary for effective adaptation?

4. Would the authors consider including suggestions for reducing computational demands in GPTBIAS, such as a streamlined scoring method?

---

### Official Review · Reviewer_U4bk · 2024-11-05

**Soundness:** 2
**Presentation:** 2
**Contribution:** 2
**Rating:** 3
**Confidence:** 4

**Summary:**

The paper introduces a bias evaluation framework that leverages advanced LLMs, such as GPT-4, to assess bias across nine distinct bias types. The framework, GPTBIAS, addresses the limitations of existing evaluation methods by providing quantitative bias scores and detailed insights into the kinds of biases, affected demographics, and suggestions for mitigation. The study emphasizes the importance of comprehensive bias assessment in the deployment of LLMs.

**Strengths:**

The research is well-structured, with a robust methodology that includes comprehensive experiments on various LLMs. The use of Bias Attack Instructions tailored for evaluating biases enhances the quality of the findings, providing nuanced insights into the biases present in different models.

The paper is written, with a logical flow that guides the reader through the framework's development, implementation, and results. Definitions of bias types and detailed explanations of the evaluation process contribute to its accessibility to a broad audience.

**Weaknesses:**

* The paper primarily focuses on English language models, which may limit the applicability of the GPTBIAS framework to non-English contexts. To improve, the authors could expand their experiments to include models trained on diverse languages and dialects, assessing the framework's effectiveness in detecting biases across different linguistic and cultural contexts.
* While the framework shows promise, it relies on patterns learned by the underlying LLM, which may not capture subtle or context-specific biases effectively. Both the proposed eval benchmark and evaluation rely on LLMs, and in the paper GPT4 specifically. This leads to biased evaluation results towards GPT4 models.
* There are multiple recent bias evaluation works, e.g. [1] which suggests a combination of multiple evaluation benchmarks may work better together rather than using a single benchmark for bias evaluation. It would be great if the authors could comment on the work and compare with the proposed framework.
*  It would be great if the authors could include more discussions and experiments about the mitigation of identified bias.

[1] Esiobu, David, et al. "ROBBIE: Robust Bias Evaluation of Large Generative Language Models." The 2023 Conference on Empirical Methods in Natural Language Processing.

**Questions:**

Please refer to "Weakness" section.

---

### Note · Authors · 2024-11-26

I have read and agree with the venue's withdrawal policy on behalf of myself and my co-authors.